# Bovine Viral Diarrhea Virus Antibody Level Variation in Newborn Calves after Vaccination of Late-Gestational Cows

**DOI:** 10.3390/vetsci10090562

**Published:** 2023-09-07

**Authors:** Ui-Hyung Kim, Sung-Sik Kang, Sun-Sik Jang, Sung Woo Kim, Ki-Yong Chung, Dong-Hun Kang, Bo-Hye Park, Seungmin Ha

**Affiliations:** 1Hanwoo Research Institute, National Institute of Animal Science, 4937, Gyeonggang-ro, Pyeongchang-gun 25340, Gangwon-do, Republic of Korea; uhkim@korea.kr (U.-H.K.); sskang84@korea.kr (S.-S.K.); jangsc@korea.kr (S.-S.J.); sungwoo@korea.kr (S.W.K.); 2Department of Beef Science, Korea National College of Agriculture and Fisheries, 1515, Kongjwipatjwi-ro, Deokjin-gu, Jeonju-si 54874, Jeollabuk-do, Republic of Korea; cky95@korea.kr (K.-Y.C.); kkongchi24@gmail.com (D.-H.K.); wodls9441@gmail.com (B.-H.P.); 3Dairy Science Division, National Institute of Animal Science, 114 Sinbang 1-gil, Seonghwan-eup, Seobuk-gu, Cheonan-si 31000, Chungcheongnam-do, Republic of Korea

**Keywords:** antibody, BVDV, calf, cow, vaccine

## Abstract

**Simple Summary:**

We confirmed that a large number of new antibodies are generated in the body of the mother after BVDV vaccination in late-stage pregnancy in beef cattle. We also observed a slow decline in BVDV maternal antibodies in calves born to pregnant cows that produced high levels of BVDV antibodies following pre-calving BVDV vaccination.

**Abstract:**

This study was conducted to confirm variation in bovine viral diarrhea virus (BVDV) antibody levels transferred to calves from their mother’s colostrum after vaccination of late-gestational cows. Blood samples were drawn from 60 pregnant cows that had been vaccinated more than one year and less than two years previously. The samples were collected six weeks prior to the expected date of delivery. After sample collection, the cows were divided into two groups of 30. One group received 2 mL of BVDV vaccine, and a control group received 2 mL of phosphate-buffered saline (PBS). Blood was collected from the cows three weeks post-administration. Additional blood samples were taken from calves at 1, 4, 8, 12, 16, and 20 weeks after birth. The serum was separated from the collected blood, and BVDV antibody changes were confirmed by enzyme-linked immunosorbent assays. BVDV antibody levels were higher from 8 to 20 weeks of age in calves born to late-gestational BVDV-vaccinated cows than in calves born to control cows (*p* < 0.0083). Further analysis confirmed a slow decline in BVDV maternal antibodies in calves born to pregnant cows that produced high levels of BVDV antibodies following pre-calving BVDV vaccination. These results suggest that BVDV vaccination of cattle in late pregnancy may help to extend the duration of protection against BVDV infection in newborn calves.

## 1. Introduction

Bovine viral diarrhea virus (BVDV) is responsible for significant losses in the cattle industry worldwide. BVDV infections in cattle may be asymptomatic or associated with severe respiratory, digestive, and reproductive clinical signs [1,2,3,4,5,6]. Infection of young calves in particular can cause various clinical symptoms, and clinical syndromes associated with the respiratory system can lead to significant production losses in the industry [7]. Calves infected with BVDV are susceptible to secondary bacterial infections and, when combined with other viral infections, may exhibit severe disease [4].

Newborn and young calves depend on passive immunity transferred from their mothers as the primary basis for protection against disease [8,9]. Serum antibodies are transferred to a newborn calf by ingestion and absorption of antibodies in the dam’s colostrum; changes in calf serum antibodies are affected by the number of antibodies ingested and absorbed [10]. Effective vaccination of pregnant cows and intake of colostrum by their calves can enhance the immune response of newborn calves [11]. Administering vaccines to cows before calving can increase specific antibodies in colostrum, and this method has been used in the administration of vaccines for *Escherichia coli*, rotavirus, and coronaviruses before calving to prevent calf diarrhea [12,13,14]. In addition, bovine herpesvirus and BVDV vaccination of late-gestational beef heifers may result in greater and more consistent deposition of specific antibodies in colostrum, reducing the variability of initial titers in calves and increasing the duration of maternal immunity [15]. Passive immunity in young calves reportedly protects against disease while interfering with a calf’s ability to develop immunity to vaccine antigens [8,9]. However, the general consensus is that a large quantity of antibodies transferred through colostrum can help prevent disease in newborn calves.

A previous study [15] discussed the transfer of BVDV antibodies from mother to calf through colostrum after vaccination in pregnant cows. However, there are few studies on subsequent changes in BVDV antibodies in calves over time. This experiment was conducted to determine whether late-gestational BVDV vaccination of pregnant cows affects the duration of BVDV maternal antibodies transferred to calves via colostrum after parturition.

## 2. Materials and Methods

### 2.1. Ethics Approval

This study was approved by the Institutional Animal Care and Use Committee of the National Institute of Animal Science, Republic of Korea (approval number: NIAS-2021-504). All experimental procedures involving animals were conducted in strict accordance with relevant guidelines and regulations. All methods used for in vivo studies in cows were in accordance with the Consolidated Standards of Reporting Trials guidelines.

### 2.2. Animals

All beef cows and calves (Hanwoo, a breed derived from *Bos taurus*) in this study were born at a farm in Daegwallyeong, Gangwondo, South Korea. All cattle raised on this farm were impregnated by artificial insemination, and the birth dates of all calves were recorded. A total of 60 healthy pregnant cows and 60 calves served as the subjects of this study. One month before delivery, all pregnant cows were moved to separate delivery stalls. Calves were housed with their mothers in separate pens for 1 week immediately after calving and suckled directly from their mothers. One week after birth, the calves were allowed access to food concentrate; at three months, hay was provided ad libitum. The calves were weaned three months after birth, and male and female calves were separated. The calves were divided into groups of 10 and transferred into pens. The calves were then fed a standard diet of concentrate, hay, and a mineral supplement. Before the experiments, all cows and calves on the farm were tested for persistent BVDV infection; no persistently infected animals were found. To monitor for persistent BVDV infection, blood was collected from all cows and calves on the farm before the experiment and tested for BVDV antigen by ELISA (IDEXX BVDV Ag/Serum Plus Test, IDEXX, Liebefeld-Bern, Switzerland); no cows and calves were found to be positive for BVDV antigen.

### 2.3. Experiment

Six weeks before the expected date of delivery, blood samples were drawn from 60 pregnant cows more than one year and less than two years after receiving BVDV vaccinations. The cows were then divided into two groups of 30. In the treatment group, 30 pregnant cows received the BVDV vaccine (CattleMaster 4). The remaining 30 cows comprised the control group to which PBS was administered. There are few studies about the antibody changes caused by this vaccine in Hanwoo. Therefore, we referred to the manual of CattleMaster 4 in this experiment. The manual recommends two doses of the vaccine 2–4 weeks apart for the primary vaccination. It was assumed that the increase in antibodies due to the first vaccination could be confirmed in the body of the cow 2–4 weeks after the first vaccination. Therefore, in this study, the antibody titer was measured 3 weeks after the vaccination to measure the change in antibodies due to the vaccination. Three weeks after vaccine or PBS administration, blood samples were taken from the cows in both groups. Blood was also sampled from calves born to the 60 cows at 1, 4, 8, 12, 16, and 20 weeks after parturition. We then compared cow and calf antibody quantities and changes over time between the vaccinated group and control group. In an additional analysis of the vaccinated group, pregnant cows were ranked according to greatest BVDV antibody increase from the day of BVDV vaccine administration to 3 weeks later. Antibody production levels of the top 10 cows and their calves (high antibody production) and the bottom 10 cows and their calves (low antibody production) were compared.

### 2.4. Vaccination

The vaccine utilized in all experiments (CattleMaster 4, Zoetis, Lincoln, NE, USA) was a freeze-dried product containing chemically altered infectious bovine rhinotracheitis and parainfluenza 3 viruses; modified live bovine respiratory syncytial virus; and a liquid, adjuvanted preparation of inactivated cytopathic and non-cytopathic bovine viral diarrhea virus. In the experimental group, 2 mL of the BVDV vaccine was administered intramuscularly; the control group received 2 mL of PBS intramuscularly. A veterinarian was present to assess the health of cows prior to administration of all vaccines.

### 2.5. Blood Sampling and BVDV Antibody Evaluation

A veterinarian also assessed the health of cows and calves at all blood-sampling sessions. Approximately 10 mL of blood was collected from the jugular vein of each cow in Vacutainer tubes (BD, Franklin Lakes, NJ, USA). Blood samples were centrifuged to collect serum samples, which were stored at −80 °C until BVDV antibody enzyme-linked immunosorbent assay (ELISA) test. Total antibodies specific for BVDV in serum samples were measured with a commercially available ELISA kit (BVDV Total Ab Test, IDEXX, Switzerland). ELISA tests were performed according to the manufacturer’s instructions. The test was performed using 1:5 serum dilution and 90 min incubation time of serum with the coated antigen. The results were expressed as sample to positive (S/P) ratio by calculating optical density (OD) values of the test samples and corrected OD values of the positive control. Samples with S/P ratio values ≥ 0.3 were classified as positive; those with values less than 0.2 were considered negative. Previous research has shown that, using the VNT as the reference standard, the ELISA manufacturer’s recommended cut-off was appropriate, with sensitivity and specificity of 96.7% and 97.1%, respectively [16,17].

### 2.6. Statistical Analysis

In the experiment, one calf from the vaccinated group and two calves from the control group were excluded from analysis due to diarrhea. Statistical analyses were performed using the Statistical Package for the Social Sciences software (version 26.0; IBM Corp., Armonk, NY, USA). The Shapiro–Wilk and Levene’s tests were used for normality analysis and equality of variances. The Mann–Whitney U with Bonferroni’s method was used to evaluate age, parity, and BVDV antibodies of dams. The chi-square test was used to determine the associations between vaccination and control groups. *p*-values < 0.05 indicated statistical significance. A generalized linear mixed model with Bonferroni post hoc analysis was performed to evaluate repeated reactivity measurements of BVDV antibodies of calves. In the generalized linear mixed model with Bonferroni post hoc analysis, time and classification (vaccination and control) were the fixed effects, whereas calves nested within the classification were the random effects. After identifying an interaction effect of classification and time, we applied the Mann–Whitney U test with Bonferroni’s correction to evaluate BVDV antibodies of calves. Results with *p*-values < 0.0083 were considered statistically significant.

## 3. Results

A comparison of the changes in cow and calf antibody quantities between the vaccinated group (29 cows) and control group (28 cows) is shown in Table 1. There was no difference in BVDV antibodies between the vaccinated and control cow groups at the time of administration. However, 3 weeks after administration, BVDV antibodies in the vaccinated group were higher than those in the control group (*p* = 0.008). There was no difference in BVDV antibody levels between calves from vaccinated and control groups between 1 and 4 weeks of age. However, BVDV antibodies in the control group declined rapidly, dropping to 0.3 by week 20. From 8 to 20 weeks of age, the BVDV antibodies of the vaccinated group calves were higher than those of the control group calves (*p* < 0.0083).

In the vaccinated group, cows were ranked according to BVDV antibody increase from the day of BVDV vaccine administration to 3 weeks post-vaccination. Based on this ranking, antibody-level changes in the top 10 cows and their calves and the bottom 10 cows and their calves were compared (Table 2). At the time of BVDV vaccine administration, BVDV antibody levels of the low antibody production cow group were higher than those of the high antibody production cow group (*p* < 0.001). However, 3 weeks after BVDV vaccination, there was no difference between the two groups. Among calves in the two groups, there was no difference in antibody levels from 1 to 16 weeks of age. However, at 20 weeks of age, BVDV antibody levels in calves born to the high antibody production cows were higher than those in calves born to low antibody production cows (*p* = 0.005).

## 4. Discussion

In this study, we investigated variation in BVDV antibodies in pregnant cows after BVDV vaccination in late gestation and confirmed post-natal changes in BVDV antibody levels in their calves.

In this experiment, there was no difference in BVDV antibody levels between the vaccinated and control groups 6 weeks prior to delivery when BVDV vaccine or PBS was administered. However, 3 weeks after vaccination, BVDV antibody levels increased in the group that received late-gestational vaccination and were higher than those in the control group. These results indicate that BVDV antibodies were effectively increased when a BVDV vaccine was administered late in pregnancy. In a previous study [15], the increase in BVDV post-vaccination antibody levels in pregnant cows at 6.5 to 8 months of gestation was similar to our findings. However, that study [15] reported that calves born to cows given a BVDV vaccine late in gestation had higher levels of BVDV antibodies 24 h after parturition compared with calves born to cows that did not undergo late-gestation vaccination. In our study, the first calf-blood collection was performed 1 week, not 24 h, post-parturition; direct comparison of results between the two studies is therefore difficult. However, the previous study [15] showed that late-gestation vaccination of beef cows resulted in greater and more consistent deposition of specific antibodies in colostrum, reducing the variability in initial specific antibody titers in calves and increasing the duration of maternal antibody transfer. In our study, the variability of initial BVDV antibody levels in calves born to pregnant cows vaccinated in late pregnancy also tended to be lower than in calves born to control-group dams. Although our experiment only measured BVDV antibodies in calves up to 20 weeks after parturition, BVDV antibodies in calves born to late-gestational vaccinated cows decreased more slowly than those in calves born to the control group. According to these results, BVDV vaccination in late pregnancy can prolong the duration of maternal BVDV antibodies transferred to newborn calves through colostrum.

According to our results, while there was no difference in BVDV antibodies between the two groups of calves at 1 and 4 weeks, there was a difference between 8 and 20 weeks of age. When BVDV antibody data for each individual was examined more closely, at the time of BVDV vaccine and PBS administration, there were 0 cows in the vaccinated group with antibody titers greater than 1.5 compared to 8 cows in the control group. The antibodies of these eight control-group cows and their calves changed as follows: in cows, 1.76 ± 0.12 at maternal PBS administration, 1.73 ± 0.10 3 weeks after administration; in calves, 2.13 ± 0.07 at 1 week, 2.00 ± 0.07 at 4 weeks, 1.71 ± 0.08 at 8 weeks, 1.49 ± 0.15 at 12 weeks, 1.15 ± 0.21 at 16 weeks, and 0.73 ± 0.23 at 20 weeks. The BVDV antibodies in the calves of these eight cows tended to remain higher than the mean in the control group calves throughout the period. Previous studies [18,19] reported that calves born to younger mothers received fewer maternal antibodies through colostrum. The average age of the eight cows with antibody levels greater than 1.5 at the time of PBS administration was 7.8 years, higher than the average age of the other cows in the control group, 3.4 years. However, the BVDV antibodies in the calves of these eight cows were compared to the mean of the vaccinated group calves and were higher at week 1 and then rapidly decreased to be lower at week 20. According to these results, in the control group, the amount of maternal colostrum and the level of BVDV antibodies in colostrum may have influenced the BVDV antibody changes of the control group calves. However, these factors are estimated to have little impact on the results related to the comparison of changes in BVDV antibodies in the body of vaccinated group calves and control group calves.

A previous study [20] reported that the duration of maternal immunity to respiratory viruses, including BVDV, was directly proportional to the serum levels of antibodies ingested and absorbed through colostrum and that calves with lower initial antibody levels became seronegative earlier in life. Compared with the results of previously reported studies, our results are difficult to interpret. Therefore, we performed several analyses to determine the cause of the slow decline in calf BVDV antibodies in the vaccinated group and found an interesting result. In the vaccinated group, cows were ranked according to highest BVDV antibody increase from the day of BVDV vaccine administration to 3 weeks later. Then, BVDV antibody changes in the top 10 vaccinated cows and their calves were compared with the bottom 10 vaccinated cows and their calves. The results of this comparison are presented in Table 2. High antibody production cows had a low BVDV antibody titer at the time of BVDV vaccination, while low antibody production cows had a high BVDV antibody titer at the time of BVDV vaccination. At three weeks post-vaccination, there was no difference between the two groups. Changes in BVDV antibody levels in calves born to these two groups of pregnant cows were compared. BVDV antibodies in the two groups of calves did not differ from week 1 to week 16; however, BVDV antibodies decreased more rapidly in the calves of low BVDV antibody production cows than in the calves of high BVDV antibody production cows. At 20 weeks of age, the BVDV antibodies in calves born to high BVDV antibody production cows were higher than in calves born to low BVDV antibody production cows. Our results suggest that BVDV antibodies are newly formed in the mother’s body before delivery and are maintained for a long time in the calf when they are transferred through colostrum. In addition, they suggest that an increase in the amount of BVDV antibodies produced in the mother’s body before delivery will increase the time maintained in the body of the calf. 

In our study, we confirmed that the duration of BVDV antibodies transferred through colostrum to calves was affected not only by the amount of BVDV antibodies ingested and absorbed through colostrum, but also by the amount of newly produced BVDV antibodies in cows before parturition.

## 5. Conclusions

We confirmed that a large number of new antibodies are generated in the body of pregnant cows after BVDV vaccination in late gestation. These antibodies were transferred to calves through colostrum and maintained for a long period of time in the body of the calf. This study confirms antibody changes due to BVDV vaccination at the end of pregnancy in beef cows. Further studies may be needed to determine if these results hold true for other types of cattle or different types of vaccines. In addition, studies related to the timing of the first BVDV vaccination of calves are needed if the mother is vaccinated during late pregnancy.

## Figures and Tables

**Table 1 vetsci-10-00562-t001:** Sample-to-positive (S/P) ratios (mean ± STD) of BVDV of vaccinated group and control group.

Variable	Vaccinated	Control	*p* Value
Number	29	28	
Dam			
Age (years)	4.55 ± 1.97	4.61 ± 2.25	0.692
Parity	2.07 ± 1.60	2.25 ± 1.62	0.820
−6 weeks	0.85 ± 0.38	0.91 ± 0.60	0.838
−3 weeks	1.41 ± 0.20 *	0.90 ± 0.58 *	0.008
Calf			
1 week	1.49 ± 0.13	1.49 ± 0.45	0.572
4 weeks	1.41 ± 0.14	1.25 ± 0.54	0.048
8 weeks	1.32 ± 0.17 ^#^	0.95 ± 0.53 ^#^	0.007
12 weeks	1.23 ± 0.21 ^#^	0.70 ± 0.54 ^#^	0.001
16 weeks	1.14 ± 0.25 ^#^	0.50 ± 0.45 ^#^	<0.001
20 weeks	1.04 ± 0.32 ^#^	0.30 ± 0.31 ^#^	<0.001

“−6 weeks” indicates 6 weeks before the expected delivery date of pregnant cows, and “−3 weeks” represents 3 weeks after BVDV vaccination and PBS inoculation. The variables 1 week, 4 weeks, 8 weeks, 12 weeks, 16 weeks, and 20 weeks represent the ages of the calves. Data are expressed as mean ± standard deviation and were compared using independent *t*-tests. An * in the same row indicates statistically significant difference at *p* < 0.05 (Mann–Whitney U test with Bonferroni’s correction). A ^#^ in the same row indicate a significant difference at *p* < 0.0083 (Mann–Whitney U test with Bonferroni’s correction).

**Table 2 vetsci-10-00562-t002:** Sample-to-positive (S/P) ratios (mean ± STD) of BVDV of the low antibody production group and high antibody production group. In the vaccinated group, pregnant cows were ranked according to greatest BVDV antibody increase from the day of BVDV vaccine administration to 3 weeks later. Antibody production levels of the top 10 cows and their calves (high antibody production) and the bottom 10 cows and their calves (low antibody production) were compared.

Variable	Low Antibody Production Cows	High Antibody Production Cows	*p* Value
Number	10	10	
Dam			
Age (years)	6.80 ± 1.48 *	3.40 ± 1.08 *	<0.001
Parity	3.60 ± 0.84 *	1.20 ± 0.42 *	0.001
−6 weeks	1.19 ± 0.25 *	0.43 ± 0.12 *	<0.001
−3 weeks	1.26 ± 0.26	1.45 ± 0.11	0.190
Calf			
1 week	1.42 ± 0.16	1.52 ± 0.13	0.481
4 weeks	1.32 ± 0.17	1.45 ± 0.13	0.165
8 weeks	1.22 ± 0.21	1.38 ± 0.14	0.123
12 weeks	1.08 ± 0.26	1.30 ± 0.13	0.052
16 weeks	0.93 ± 0.29	1.27 ± 0.15	0.011
20 weeks	0.74 ± 0.35 ^#^	1.20 ± 0.18 ^#^	0.005

“−6 weeks” indicates 6 weeks before the expected delivery date of pregnant cows, and “−3 weeks” represents 3 weeks after BVDV vaccination and PBS inoculation. The variables 1 week, 4 weeks, 8 weeks, 12 weeks, 16 weeks, and 20 weeks represent the ages of the calves. Data are expressed as mean ± standard deviation and were compared using independent *t*-tests. An * in the same row indicates statistically significant difference at *p* < 0.05 (Mann–Whitney U test with Bonferroni’s correction). A ^#^ in the same row indicate a significant difference at *p* < 0.0083 (Mann–Whitney U test with Bonferroni’s correction).

## Data Availability

The data presented in this study are available on request from the corresponding author.

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
