# Peer review of "Bovine Viral Diarrhea Virus Antibody Level Variation in Newborn Calves after Vaccination of Late-Gestational Cows"

_vetsci, 2023, doi:10.3390/vetsci10090562_

Round 1

Reviewer 1 Report

Reviewer comments for manuscript ID vetsci- 2589135 entitled ‘Bovine viral diarrhea virus antibody level variation in newborn calves after vaccination of late-gestational cows’

General Comments

Bovine viral diarrhoea is an economically significant disease for the dairy industry. Calf mortality is the significant factor in the economics of dairy enterprises. Calfhood vaccination is the general practice against various viral and bacterial diseases. However, the efficacy of vaccination at this stage is variable. Vaccination of advance pregnant dams can provide passive immunity of the newborn calves that can further be strengthened by vaccinating them. This strategy can reduce calf mortality and morbidity due to many diseases.

It is a nice attempt by the authors to research an important disease in calves. The manuscript is well written with a flawless writing. The statistical analysis is relevant and robust. Results are nicely presented. However, the discussion is weak as it merely describes the results. Indepth analysis of the result is lacking. I would like the authors to work on this particular aspect of the document.

Specific comments

Line99-101: Please clarify the basis of this schedule of measuring antibody titres – at 3 weeks after vaccination.

Lines 207-09: Why BVDV antibodies in calves born to late-gestational vaccinated cows decreased more slowly than those in calves born to the control group in your study? Please clarify.

Lines 221-26: Can the amount of colostrum ingested by the calves and the level of antibodies in the colostrum of individual dams be the confounders in this study? Please clarify.

Author Response

Thank you for your review of the paper.

Line99-101: Please clarify the basis of this schedule of measuring antibody titres – at 3 weeks after vaccination.

There were few research about the antibody changes caused by the use of this vaccine (cattlemaster4) in Korean beef cattle. Therefore, we referred to the manual of Cattlmaster4 used in this experiment. The manual recommends two doses of the vaccine 2-4 weeks apart for the primary vaccination. Therefore, it was assumed that the increase in antibodies due to the first vaccination could be confirmed in the body of the individual 2-4 weeks after the first vaccination. Therefore, in this study, the antibody titer was measured 3 weeks after the vaccination to check the change in antibodies due to the vaccination, assuming that it was revaccination instead of a primary vaccination.

Lines 207-09: Why BVDV antibodies in calves born to late-gestational vaccinated cows decreased more slowly than those in calves born to the control group in your study? Please clarify.

The points you mention are addressed at the end of the Discussion section of this paper. The reason for the slow decline in BVDV antibodies in calves born to late-gestational vaccinated cows, is thought to be that the newly produced antibodies from the mothers' late pregnancy vaccination persisted long after transfer to the colostrum. Although not directly addressed in this study, it is thought that antibodies have a lifespan. It is thought that the antibodies with a short lifespan will decline rapidly, even if you have a lot of them. It is thought that new, new produced, long lifespan antibodies may have a greater impact on antibody persistence in calves than large amounts of antibodies. Therefore, it is thought that further research on the lifespan of vaccine antibodies is needed.

Lines 221-26: Can the amount of colostrum ingested by the calves and the level of antibodies in the colostrum of individual dams be the confounders in this study? Please clarify.

Among our studies, there are additional analysis results that are not presented in the main text. In the control group, pregnant cows were ranked based on BVDV antibodies measured at 3 weeks post-administration of PBS. The BVDV antibody levels of the top 10 cows were compared to those of the bottom 10 cows. It was confirmed that BVDV antibodies in calves born in the top 10 cows lasted longer than BVDV antibodies in calves born in the bottom 10 cows. Previous study (Environmental and management factors influencing BVDV antibody levels and response to vaccination in weanling calves. Proceedings Beef Improvement Federation 43rd Annual Research Symposium and Annual Meeting, 32-4) reported that calves born to younger mothers received fewer maternal antibodies through colostrum. We looked at the age of these pregnant cows of the analysis and found that the top 10 cows were 7 years old and the bottom 10 cows were 3 years old. I suspect that older pregnant cows may have more antibodies in their colostrum and a higher volume of colostrum. And these results are similar to previous studies. In addition, it was confirmed that the higher the initial BVDV antibody titer in the calf, the longer it persists in the calf's body.

In the control group mentioned in our papers discussion, the age of the eight cows with antibodies above 1.5 at the time of PBS vaccination was 7 years. However, vaccine antibodies in calves born to these pregnant cows declined rapidly from 2.13 at 1 week to 0.73 at 20 weeks. This was lower than the 20-week 1.04 in the vaccinated group calves. According to these results, in the control group, the amount of maternal colostrum and the level of antibodies in colostrum may have influenced the result of the comparison of top 10 cows and bottom 10 cows. But are not believed to have influenced the outcome related to the decrease in maternal antibodies in the bodies of vaccinated group and control group calves.

Reviewer 2 Report

Here the authors analyzed the antibody level against Bovine viral diarrhea virus in newborn calves. They found that the antibody could consistently maintain higher level in newborn calves from vaccinated cattle. While it is basic found without surprising, the results were important to guide the vaccination strategy.

This is a well-designed and well-written study that may broad implications for the understanding of immune response with BVDV vaccination. I thought the data was rigorous and a good story that is suitable for this journal.

The remaining important question is the neutralization antibody. The authors observed higher antibody in newborn calves from vaccination group. Does it provide better neutralization effect against BVDV infection? They could try it in cell culture system or discuss it.

Author Response

Thank you for your review of the paper.

Similar to your comments, we believe that a VNT analysis would have provided more conclusive results, so we regret that we were unable to present VNT results in this study.

However, previous research has shown that the using the VNT as the reference standard, the ELISA manufacturer's recommended cut-off was confirmed as appropriate, and the sensitivity and specificity of the ELISA were found to be 96.7% and 97.1%, respectively, compared with the VNT ( Lanyon, S.R.; Anderson, M.L.; Bergman, E.; Reichel, M.P. Validation and evaluation of a commercially available ELISA for the detection of antibodies specific to bovine viral diarrhoea virus (bovine pestivirus). Aust. Vet. J. 2013, 91, 52-56).

Therefore, we believe that the ELISA results in this study are sufficiently reliable.